# Development of a Dragonfly-Inspired High Aerodynamic Force Flapping-Wing Mechanism Using Asymmetric Wing Flapping Motion

**DOI:** 10.3390/biomimetics10050309

**Published:** 2025-05-11

**Authors:** Jinze Liang, Mengzong Zheng, Tianyu Pan, Guanting Su, Yuanjun Deng, Mengda Cao, Qiushi Li

**Affiliations:** 1Research Institute of Aero-Engine, Beihang University, Beijing 100191, China; liangjinze@buaa.edu.cn (J.L.); zhengmengzong@buaa.edu.cn (M.Z.); pantianyu@buaa.edu.cn (T.P.); dengyuanjun@buaa.edu.cn (Y.D.); ZY2232149@buaa.edu.cn (M.C.); liqs@buaa.edu.cn (Q.L.); 2National Key Laboratory of Science and Technology on Aero-Engine Aero-thermodynamics, Beihang University, Beijing 100191, China; 3Beihang (Sichuan) Western International Innovation Port Technology Co., Ltd., Chengdu 610299, China; 4School of Energy and Power Engineering, Beihang University, Beijing 100191, China; 5Key Laboratory of Fluid and Power Machinery, Ministry of Education, Xihua University, Chengdu 610039, China; 6Engineering Research Center of Intelligent Air-Ground Integration Vehicle and Control, Ministry of Education, Xihua University, Chengdu 610039, China

**Keywords:** dragonfly, biomimetic mechanism design, two degrees of freedom, asymmetric motion, aerodynamic analysis

## Abstract

Bionic micro air vehicles are currently being popularized for military as well as civilian use and dragonflies display a wealth of skill in their remarkable flight capabilities. This study designs an asymmetric motion flapping-wing mechanism inspired by the dragonfly, using a single actuator to achieve the coupling of stroke and pitch motion. This study simulates the motion of the dragonfly’s wings using the designed mechanism and experimentally validates the motion laws and aerodynamic characteristics of the mechanism. The analysis focuses on the asymmetry in the wing’s stroke and pitch motion and their aerodynamic implications. The flapping-wing mechanism accurately replicates the wing motion of a real dragonfly in flight, and the maximum lift-to-weight ratio can reach up to 230.2%, demonstrating significant aerodynamic benefits. This mechanism provides valuable guidance for the structural design and kinematic control of future flapping-wing vehicles.

## 1. Introduction

Bionic flapping-wing micro air vehicles (MAVs) mimic the flapping motion of insects, an approach which shows great potential in areas such as reconnaissance, environmental monitoring, military operations, and disaster relief [1]. Compared with fixed-wing and rotary-wing aircraft, the flapping-wing mode produces lift utilizing unsteady aerodynamic effects. The lift mechanism in flapping flight is complex, involving not only the traditional lift generated by the downward motion of the wings driving air during the downstroke but also the vortex effects during the flapping process, including the contribution of the leading-edge vortex and wingtip vortices [2,3].

In nature, the flapping movement of insects can be described by three movements: stroke, pitch and deviation. The stroke and pitch movements are mainly related to the cause of aerodynamic force, while the deviation movement is mainly related to the distribution of aerodynamic torque [4]. Researchers have identified the pitch motion as an important unsteady mechanism for generating high lift in flapping flight. Kramer [5] experimentally observed that insect wings rotate about their axis at the end of each downstroke, ensuring that the wings maintain a positive angle of attack throughout both upstroke and downstroke. Through an experimental study on fruit fly wing motion, Dickinson [6] discovered that the wings generate significant unsteady high lift during the beginning and end phases of the flapping cycle. The wings rapidly rotate, causing the leading-edge vortex (LEV) and trailing-edge vortex (TEV) to shed, after which the wings enter the reverse stroke phase, interacting with the shed vortices to generate higher aerodynamic forces. This phenomenon is known as the “wake capture” effect. Research indicates that when pitch is present, the lift produced by flapping wings increases several times compared to that produced by a simple stroke motion. Therefore, studying the pitch mechanism of flapping is crucial for improving the aerodynamic performance of flapping-wing vehicles.

Most of the existing bionic insect flapping wing mechanisms can be divided into two categories: the first category is a single-drive single-degree-of-freedom flapping wing mechanism that can achieve control only on stroke motion. For pitch motion, most mechanisms passively adjust by aerodynamic force acting on the flexible wing to deform it. The “Microbat” developed by the California Institute of Technology [7] was the first flapping-wing aircraft to use a micro-motor as a driving element, pioneering the integration of micro-electromechanical systems (MEMS) into flapping-wing prototypes. Park et al. [8] successfully developed a tail-less insect-inspired FMAV by mimicking beetles, which used a crank-slider mechanism to convert the motor’s pitch motion into wing flapping. Roshanbin [9] developed a motor-driven flapping-wing aircraft and proposed a “wing twist adjustment” structure, which can change the curvature and deformation of the wing surface during the stroke process, enabling stability control in two degrees of freedom (roll and pitch) of the aircraft. However, these flapping-wing aircraft use passive pitch, which fails to simulate the active pitch of real insect wings. This biomimetic flaw limits the aircraft’s ability to fully optimize its aerodynamic performance. The second category is a dual-drive two-degrees-of-freedom flapping wing mechanism that can achieve control of stroke and pitch motion simultaneously. This type of mechanism is mostly used in bionic insect flapping-wing test benches. Phillips and Bomphrey [10,11] designed a high-speed flapping mechanism in air and studied the effects of wing aspect ratio and the wing root structure on aerodynamic performance. Han et al. [12,13,14,15,16] used a towing tank to simulate the flapping motion of an eagle moth and investigated aerodynamic parameters under different advance ratios and stroke angles. Maybury and Lehmann [17,18] employed a dual-motor design for a two-degrees-of-freedom mechanical wing, simulating dragonfly wings in a water tank and measuring the flow field. The second type of flapping-wing mechanism is large in size and has redundant structure, which cannot meet the size or weight requirements of flapping-wing MAVs.

Dragonflies have superb flying abilities [19,20,21]. Biological observations have found that dragonflies use special asymmetric wing flapping motions, and the corresponding wing stroke angle and pitch angle also show significant asymmetry in the downstroke and upstroke stages [22]. The asymmetry of the motion patterns also leads to essential differences in the aerodynamic characteristics of dragonfly flapping wings in the upstroke and downstroke stages, which is significantly different from ordinary insects (such as fruit flies, hawk moths, etc.). To date, no flapping-wing mechanism with high aerodynamic performance has been developed that effectively utilizes the unique flapping patterns of dragonflies, and the underlying mechanism of its influence on aerodynamic performance is still unclear. To address the above issues, based on existing research, this study applies biomimetic principles and proposes a two-degrees-of-freedom flapping-wing mechanism driven by a single motor, inspired by dragonfly wings, with dimensions closely resembling those of a real dragonfly. By changing the rotational speed of the motor and the position of the connection point, we can get the fluttering wing motion that we need; the realization form is simple and the bionic effect is good, and it is unique in the field of fluttering wings. Finally, co-simulation using multiple software tools is conducted to investigate the aerodynamic characteristics of the proposed flapping-wing underlying mechanism and to reveal the mechanisms of flow field evolution.

## 2. Methodology

### 2.1. Design of the Flapping-Wing Mechanism

Most advanced flapping-wing mechanisms currently utilize DC motors to drive flapping motion, making it challenging to achieve control of stroke and pitch motion simultaneously. To address this issue, this study proposes a novel transmission mechanism that uses a single motor to enable the coupling of stroking and pitching motions. The mechanism for a right wing is selected for decomposition, as shown in Figure 1. 

The transmission process of the mechanism is as follows: The driving gear rotates synchronously with the motor at an angular velocity $\omega_{1}$, while the driven gear serves to reduce the speed. The center of the driven gear passes through the transmission shaft, and the driven gear is tightly coupled with the transmission shaft to achieve synchronized rotation at a speed of $\omega_{2}$. Both ends of the transmission shaft are constrained by the frame, ensuring only one degree of pitch freedom. The bottom through-holes of the two rotational axes form revolute pairs with the frame and transmission shaft, with no displacement along the Z-axis. The other ends of the two rotational axes form revolute pairs with the endpoints of Linkage 1 and Linkage 2, respectively. Linkage 1 and Linkage 2 are connected by a revolute pair, and the other end of Linkage 2 is connected to the flapping wing. One end of the flapping wing is mounted into a hole at the endpoint of Linkage 2, effectively fixing the wing to Linkage 2. Through this transmission process, the stroke motion and pitch motion of the wing are coupled, simulating the motion patterns of dragonfly wings.

The flight performance of flying animals is closely related to parameters such as body weight, flapping frequency, and wingspan, a relationship known as the scaling law [23]. Considering the current limitations of materials and manufacturing techniques, it is challenging to design a flapping-wing mechanism with the same size and weight as a real dragonfly. In this study, the proposed mechanism has a mass of 10 g, with the flapping wing area designed at 0.0037 m^2^, a wingspan of 0.12 m, and an aspect ratio of 6.56, based on the scaling law. The skeleton material adopts a carbon fiber structure, and the wing membrane uses P31N flexible film. Based on the previous observations and intensity measurement results of the dragonfly wing vein distribution by our research team [21], the structure of the flapping wing model is designed as shown in Figure 2.

### 2.2. Kinematic Parameters of Flapping Motion and Adjustment Rules

To analyze the motion of the wing, the origin of the reference coordinate system is fixed at the wing root, which effectively simplifies the analysis model. This study adopts the method proposed by Ellington [24] to define the stroke plane. The trajectory of the dragonfly’s wingtip is projected over a number of beat cycles onto the symmetry plane of the body, the OXY plane defined in the body motion parameters. The linear regression line for these projected points is obtained using the least squares method. The stroke plane of the dragonfly is then defined as the plane passing through the wing root point and parallel to this regression line, as shown by Plane A in Figure 3. Additionally, Plane B is introduced, which is perpendicular to Plane A and passes through the origin O. The stroke plane angle θ is the angle between the stroke plane and the horizontal plane. The stroke angle α is the angle between the projection of the line connecting the wingtip and root onto the stroke plane and the Z-axis. The rotation angle β is the angle between the stroke plane and the plane of the wing. The deviation angle γ is the angle between the line connecting the wingtip and root and its projection onto the stroke plane. Since the deviation angle varies irregularly and has a small value, its influence is neglected.

Most insects (e.g., bees) use a stroke plane that is parallel to the horizontal plane during flight, with the upstroke and downstroke rotation angles following the same variation pattern, referred to as symmetric rotation, as shown in Figure 4a [25]. However, during dragonfly flight, the stroke plane forms a certain angle with the horizontal plane, and the rotation angle exhibits an asymmetric variation pattern, known as asymmetric rotation, as shown in Figure 4b.

To investigate the asymmetry of the dragonfly’s wing rotation motion, this study introduces the pitch deviation angle σ to describe the asymmetric process of wing rotation during the downstroke and upstroke phases. The pitch deviation angle σ is defined as follows: Select the midpoints of the downstroke and upstroke phases within the same motion cycle, at the section positions of the flapping wing’s chord line. After connecting the leading edges, the bisector of the angle formed between the leading edge and the stroke plane’s normal direction (denoted as PQ, ‘P’Q’) is measured, as shown in Figure 5. The angle between this bisector and the normal direction of the stroke plane is defined as the deviation angle of rotation. It can be observed that for symmetric rotation, σ = 0°, while for asymmetric rotation, σ can be either positive or negative.

Once the stroke plane is determined, the deviation angle of rotation σ can be defined based on the installation angle δ between the flapping wing and Linkage 2, as shown in Figure 6. Initially, when the surface of the flapping wing is parallel to the upper surface of Linkage 2, the deviation angle of rotation σ is 0°, representing symmetric pitch motion. When the installation angle is altered such that there is an angular offset between the flapping wing surface and the upper surface of Linkage 2, the deviation angle of rotation σ becomes nonzero, σ and δ are equal. By adjusting the installation angle δ, the effect of the deviation angle of rotation σ on the aerodynamic characteristics of the flapping wing can be explored.

In natural flight, the surrounding flow field of a dragonfly constantly changes. The uncertainty in airspeed forces the dragonfly to continuously adjust its flapping motion to maintain stable flight. Observations from Norberg [26,27] and Zheng [28] on the dragonfly’s forward flight process revealed that, under the condition of constant stroke amplitude φ, the dragonfly can adapt to changes in the surrounding flow field by altering the positions of the downstroke and upstroke starting points relative to its body. To investigate the asymmetry in the dragonfly’s wing stroke motion, both the upstroke $\varphi_{up}$ and downstroke $\varphi_{down}$ amplitudes are defined. This study introduces the stroke midpoint angle ε to explore the impact of the downstroke and upstroke starting point positions on aerodynamic forces. The positions of the downstroke and upstroke starting points are simplified as the projection of the line connecting the wingtip and root onto the stroke plane (Plane A). The midpoint angle ε is defined as the angle between the bisector of the angle formed by the downstroke and upstroke starting positions and Plane B. The angle increases in the clockwise direction and is considered positive, as shown in Figure 7.

To study the influence of the stroke midpoint angle ε on the aerodynamic performance of the flapping wing, the Plane A perspective is selected, as shown in Figure 8a. The starting positions of the downstroke and upstroke are fixed, with the revolute pair formed by Linkage 1 and the rotational axis remaining unchanged, defined as point M. A local coordinate system M-Y’Z’ is established. The revolute pair formed by Linkage 2 and the rotational axis at the downstroke starting position is defined as point N, and the revolute pair formed by Linkage 1 and Linkage 2 is defined as C_1_. Similarly, the revolute pairs at the upstroke starting position are defined as N’ and C_2_. This design adopts a clever approach: by altering the positions of the revolute pairs, any combination of φ and ε can be achieved. Figure 8a can be simplified to Figure 8b, where, under the same motion mode, the distances NC_1_ and N’C_2_ are equal. Taking N and N’ as the centers and the distance NC_1_ and N’C_2_ as the radius, circles are drawn in the MY’Z’ plane. Lines $s_{1}$ and $l_{1}$ are parallel to MZ’, while $s_{2}$ and $l_{2}$ intersect the circles centered at N and N’ at C_1_ and C_2_, respectively. Another circle is drawn using M as the center and radius $r_{2}$, ensuring that both C_1_ and C_2_ lie on this circle. In this design, the coordinates are defined as follows: M (0, 0); C_1_ (z_1_, y_1_); C_2_ (z_2_, y_2_); N (1, y_3_) and N′ (1, y_4_).

As shown in Figure 8c, the rotational centers of the two rotational axes are defined as D and D’, respectively. The rotational radius of D’ around O is $r_{3}$. The line OD’ is perpendicular to DD’, and $\beta^{\prime }$ is the angle between DD’ and DO. The length of DO is 13 mm, arcsin $\beta^{\prime }$= $r_{3}$/13, $y_{3}$ = 13 − $r_{1}$, $y_{4}$ = 13 + $r_{1}$. From the geometric relationship, the maximum value of the pitch angle can be derived as 90° + $\beta^{\prime }$ + δ, and the minimum value of the pitch angle β can be derived as 90 − $\beta^{\prime }$+ δ. The remaining parameters can be obtained by solving the system of equations, as follows:(1)$$\left\{ \begin{array}{l} z_{1}^{2}+y_{1}^{2}=r_{2}^{2}\\ z_{2}^{2}+y_{2}^{2}=r_{2}^{2}\\ {\left(z_{1}-1\right)}^{2}+{\left(y_{1}-y_{3}\right)}^{2}=r_{1}^{2}\\ {\left(z_{2}-1\right)}^{2}+{\left(y_{2}-y_{4}\right)}^{2}=r_{1}^{2}\\ \phi_{down}=\arctan\frac{y_{2}-y_{4}}{z_{2}-1}\\ \phi_{up}=\arctan\frac{y_{1}-y_{3}}{z_{1}-1} \end{array}\right.$$

### 2.3. Kinematic Analysis of the Mechanism

By ensuring similarity in dimensionless parameters between the motion of the model wing and the dragonfly wing, the experimental model can accurately replicate the real flow field of a dragonfly’s flapping wings. The Reynolds number (*Re*) is a dimensionless parameter that characterizes the flow conditions of a fluid. To ensure that the flow field generated by the model wing in the experiment matches the flow field of a real dragonfly’s flapping wings, the experimental parameters of the model must be designed to maintain consistency in Re. In this study, the Reynolds number is defined as follows:(2)$$Re=\frac{\rho U_{ref}L_{ref}}{\mu },$$
where ρ represents the fluid density, μ is the fluid dynamic viscosity coefficient, $U_{ref}$ is the reference velocity, and $L_{ref}$ is the characteristic length. The chord length c of the flapping wing model is chosen as $L_{ref}$. For $U_{ref}$, the average flapping tip velocity is generally used as the reference velocity.

Meanwhile, the Strouhal number ($S_{t}$) is a similarity criterion characterizing the unsteady flow and is the similarity criterion to be simulated in unsteady aerodynamic experiments. For periodic unsteady flow, the Strouhal number is defined as follows:(3)$$St=\frac{fL_{ref}}{U_{ref}}$$
where f represents flapping frequency. At present, most scholars, when studying the upstroke and downstroke movements of dragonflies, assume that the durations of the two stages are the same for simplicity [29,30]. Based on data observed in real dragonfly flight, the standard motion parameters are given as follows: θ = 60°, φ = 60°, ε = −10°, and the pitch amplitude φ’ is within the range of 45–70°, σ = −5° [31,32]. Based on these definitions, aerodynamic analysis is conducted in Adams (2019, 64bit) , with the right wingtip marker defined as Marker-tip, as shown in Figure 9a. To better observe the variation of aerodynamic forces during the cycle, the time *T* is defined in a dimensionless form. It is specified as the time (t) after the start of the wing’s flapping motion, normalized relative to the duration of a full motion cycle. For example, *T* = 0–0.5 represents the downstroke phase, while *T* = 0.5–1 represents the upstroke phase.

The “8”-shaped flapping of dragonfly wings is a sophisticated adaptive strategy formed over billions of years of evolution. Du and Sun [33] showed that dragonflies follow an “8”-shaped movement pattern. During the downstroke stage of their wings, the movement of the flapping wings is almost parallel to the direction of gravity, which can generate a large lift force to overcome gravity and fly. In the upstroke stage, the movement of the flapping wings is almost parallel to the horizontal plane, which can generate a large thrust force to make them fly forward. Through vortex control, energy recovery and movement flexibility, it perfectly balances lift demand and energy consumption. This mechanism not only reveals the physical nature of biological flight, but also provides important inspiration for human engineering. Through Adams motion simulation, the wingtip trajectory of the flapping-wing mechanism is obtained, as shown in Figure 9b. From the simulation results in Figure 9b, it can be seen that the wingtip trajectory forms an approximately three-dimensional “8” shape, which is consistent with the real flapping motion trajectory, indicating that the model exhibits good biomimetic performance. To ensure Reynolds number similarity, it is necessary to determine the flapping frequency. Under the specified motion pattern, a stable motion cycle is selected, and the flapping frequency f is set between 5 and 20 Hz. The velocity curve of the Marker-tip is shown in Figure 9c. Using Equations (2) and (3), *Re* of the flapping motion at 20 Hz is calculated as 1648, and $S_{t}$ is calculated as 0.13. In actual observations, *Re* of dragonfly flight ranges from 1000 to 2000, and $S_{t}$ of dragonfly flight ranges from 0.11 to 0.15 [34]. Therefore, under this motion pattern, *Re* and $S_{t}$ of the flapping wing matches the observed values, ensuring that the flow field generated by the model flapping wing is similar to that of a real dragonfly. Additionally, the motion curves of the stroke angle α and pitch angle β are calculated, as shown in Figure 9d. The variation patterns of α and β align with the set parameters, verifying the rationality of the mechanism design.

### 2.4. Experimental Measurement

To reduce weight and ensure structural rigidity, most parts in this design are manufactured using PE material, while the gears are 3D-printed from plastic. The flapping wings use P31N thin film as the membrane, with carbon fiber rods serving as the framework, fixed at the joints with adhesive. The connection process primarily employs pin connections and mechanical fasteners. The control system uses a custom-designed integrated circuit chip, including a motor drive module, a receiver serial port module, and a voltage module. The control chip operates at a maximum voltage of 7.4 V and a maximum current of 2 A, with the entire system powered by a stable 3.7 V DC power source. A schematic diagram of the control system is shown in Figure 10a. Based on the simulation results of the flapping mechanism discussed earlier, this study employs a hollow-cup motor with a mass of only 2 g, a no-load speed of 48,000 r/min, a no-load current of 70 mA, and a rated voltage of 3.7 V. Examples of the parts are shown in Figure 10b, and the total weight of the flapping-wing mechanism is 10.2 g, as shown in Figure 10c. To avoid interference in the flapping wing’s flow field caused by environmental changes, the motor’s start and stop operations are controlled via a remote controller that sends signals to the receiver.

In this experiment, a specific model of a three-dimensional force sensor with a measurement range of −50 N to 50 N was used. Before conducting the experiment, the sensor was calibrated by applying known loads to establish the relationship between the displayed voltage values and the actual applied forces. Table 1 presents the calibration data of the sensor.

The relationship between the loading output voltage and the pressure values along the three axes is derived from the table, as follows:(4)$$V=-\frac{1}{4}F,$$
where V represents the actual displayed voltage (V), and F represents the measured force (N). When the sensor is under tension, both the standard pressure value and the loaded output voltage are positive, but the linear relationship remains unchanged. Using the derived calibration formula and the actual measured voltage values along the X-axis and Y-axis, the thrust and lift on the X-axis and Y-axis are calculated. The error estimation in this experiment is performed at a 95% confidence level, and the error consists of two components: accuracy error P and offset error B [35,36]. The main error in this experiment is the offset error. Assuming that the measurement variables $X_{i}$ have M offset error limit, the offset error limit for the variable $X_{i}$ is calculated as the root mean square of each offset error limit $B_{i}$:(5)$$B_{i}={\left[\sum_{k=1}^{M}{\left(B_{i}\right)}_{k}^{2}\right]}^{\frac{1}{2}}$$

Based on the previous calculation criteria of the experimental bench error [28], in this experiment, the offset error consists of the following components: the stroke plane angle, the stroke angle, the pitch angle, the sensor measurement accuracy, data acquisition system, and data processing errors. The offset error of the stroke plane angle is caused by the resolution of the angle scale during the installation and calibration process, with a precision of 1°. Therefore, the offset error for the stroke plane angle is 1°/360° = 0.278%. For the data acquisition system, the system resolution used in this experiment is 16 bits, resulting in an error of 1/2^16^ = 0.0015%. The offset error from the data processing system is due to rounding errors during numerical calculations, with the software used in this experiment achieving a precision of 64 bits. The rounding error of 1/2^64^ = 10^−20^% can be neglected. The accuracy of the transmission mechanism control is maintained at 0.1% for the measurements in the X’, Y’, and Z’ directions. Additionally, the offset error for the stroke angle is 0.98%, and that for the pitch angle is 0.75%. During the experiment, the ambient temperature variation was within 20 °C ± 2 °C. Based on the above information and the definition of offset error, the total offset error of the system is determined to be 1.277%. The previous offset error of the force measurement platform of our team was 1.43% [28]. Currently, if the offset error of the force measurement platform is controlled within 5%, it can be considered that the accuracy of the force measurement platform has reached a relatively advanced level [37]. Therefore, the uncertainty of the experimental setup is confirmed to meet the requirements of the experiment. The flapping-wing mechanism was assembled and mounted onto the force measurement platform, with motion images captured during the process. When fixing the flapping wing, care was taken to align the center of mass of the flapping mechanism with the center of the three-dimensional force sensor along the vertical direction. This alignment was intended to ensure that the measured lift varied along the vertical direction, and the thrust varied along the horizontal direction, minimizing errors caused by the torque generated due to the offset of the mechanism’s center of mass. This setup ensured that both the flapping-wing prototype and the three-dimensional force sensor were securely fixed on the test platform. The fixation of the three-dimensional force sensor and the flapping mechanism is shown in Figure 11. Here, the Y-axis is the vertical direction, with the positive Y-axis pointing downward. When the voltage output along the Y-axis is negative, it indicates that the flapping wing has generated positive lift.

When designing the flapping-wing mechanism, lift is often the primary focus for researchers. Therefore, this experiment primarily investigates the variation pattern of lift during the experimental process. To preliminarily explore the impact of the pitch deviation angle σ and the stroke midpoint angle ε on the aerodynamic forces of the flapping wing, mechanical tests were conducted on the motion of a single wing of the flapping mechanism. According to the designed mechanism dimensions, the adjustment ranges of both σ and ε are −10° to 10°. Therefore, −10°, 0°, and 10° are selected as experimental parameters to verify the aerodynamic force variation law at the limit position. Meanwhile, in this study, 5° difference was also selected as an experimental parameter for analysis, and it was found that the obtained aerodynamic characteristic laws were similar to that at 10° difference, and the laws were more obvious when changing at 10° difference, so 10° difference was selected as the experimental parameter. Two sets of experiments were designed, and the parameters for both experimental groups are listed in Table 2.

### 2.5. Simulation Setup

The experimental results obtained can measure the aerodynamic performance of the flapping-wing mechanism on a macro scale. To explore the deeper principles, numerical simulation is needed to analyze the flow field structure. At the same time, the results of numerical simulation can guide the construction of the actual model. The fluid motion around the flapping wing is a low Reynolds number flow. This paper adopts a smooth rigid wing model and focuses on the unsteady aerodynamic mechanism caused by the asymmetric motion of the flapping wing, ignoring the development of smaller-scale vortices caused by the wing structure [38]. Therefore, in the subsequent analysis, the focus is on the influence of the LEV and the TEV on the unsteady aerodynamic force in the flapping-wing motion. This paper uses Adams 16.0 and XFlow 2020 for collaborative simulation to analyze the aerodynamic characteristics of the bionic flapping-wing aircraft. Through interface data transmission, repeated definition and programming can be avoided, the calculation accuracy can be improved, and time and manpower can be saved [39]. XFlow is a commercial fluid simulation software which uses the lattice Boltzmann method [40,41,42,43].

To avoid changes in the motion pattern due to flexible deformation, the influence of a single variable is studied, and the flapping wing is set to be rigid. To eliminate external environmental influences on the aerodynamic characteristics of the flapping wing, adaptive trajectory improvement and wake enhancement are employed in the simulation. The medium of the virtual wind tunnel is set to air, with a flow velocity of 0 m/s, a temperature of 289.15 K, fluid density of 1.23 kg/m^3^, coefficient of dynamic viscosity of 1.79 × 10^−5^ pa∙s, gravitational acceleration of −9.81 m/s^2^, and the wind tunnel outlet pressure set to 0 pa. The wall-relative slip is set to 0. When determining the computational domain and boundary conditions, the virtual wind tunnel is chosen as the computational domain type, and its size is optimized. A computational domain that is too small would limit the aerodynamic influence of the flapping wing, resulting in inaccurate results, while an excessively large domain would reduce simulation accuracy and increase computation time. After a comprehensive evaluation, only the right-wing model was retained in the simulation model to avoid wall effects.

In the previous work by our team, the accuracy verification and grid independence verification of the numerical simulation method were completed [21]. The computational domain is set as 30c × 30c × 30c (where c is the average chord length of the flapping wing), and the location of the computational domain and the flapping wing is shown in Figure 12a. The fluid domain mesh setup is shown in Figure 12b.

In the study of the aerodynamic characteristics of the flapping wing, the primary focus is on analyzing the variation of lift and thrust. Lift $F_{Y}$ and thrust $F_{X}$ are defined as the aerodynamic forces perpendicular to and parallel to the horizontal plane, respectively. For comparing lift and thrust under different conditions, dimensionless coefficients such as the lift coefficient $C_{L}$ and thrust coefficient $C_{T}$ are commonly used. The lift coefficient and thrust coefficient are defined as follows:(6)$$\left\{ \begin{array}{l} C_{L}=\frac{F_{Y}}{0.5\rho SU_{ref}}\\ C_{T}=\frac{F_{X}}{0.5\rho SU_{ref}} \end{array}\right.$$
where ρ represents the fluid density, S is the surface area of the flapping wing model, and $U_{ref}$ is the average flapping tip velocity. In the following, different parameter values are set for the mechanism, and dynamic analysis and fluid flow simulations are conducted to evaluate the effects of different pitch deviation angles σ, the stroke midpoints ε, and pitch amplitude ranges $\varphi^{\prime }$ ($\beta_{max}-\beta_{min}$) on the aerodynamic characteristics of the bionic flapping-wing mechanism. A total of 13 sets of experiments are designed, with the parameter values defined as shown in Table 3.

## 3. Results and Discussion

By analyzing the aforementioned experimental and numerical methods, Section 3.1 provides the experimental results, decomposes the kinematics of the flapping wing mechanism designed in this paper and shows that it has high aerodynamic performance. Section 3.2, Section 3.3 and Section 3.4 is the simulation results, and explores the underlying mechanism of the asymmetric motion of the flapping wing affecting the aerodynamic characteristics.

### 3.1. Analysis of Experimental Results

The motion of the flapping wing during a single cycle was captured and is shown in Figure 13 (See related video in Appendix A). It is clearly visible that the motion of the wingtip follows an “8”-shaped trajectory, consistent with the designed motion path. During the downstroke phase, the wing is almost vertical to the horizontal plane. In the upstroke phase, the wing is parallel to the horizontal plane, and clear pitch movement occurs at the endpoints of both the upstroke and downstroke.

Data from a complete cycle in the stable flow field for each experimental group were selected, and after processing, the lift curves were obtained, as shown in Figure 14. From Figure 14, it can be observed that, compared to σ = 0°, reducing σ is detrimental to the generation of the maximum lift, but it can significantly increase the minimum lift. Increasing σ tends to produce unfavorable effects on the extreme values of lift. It can also be observed that, compared to ε = 0°, increasing ε increases the extreme values of lift. However, the lift curve at ε = −10° is mostly below that at ε = 0° for the entire cycle, which is unfavorable for lift generation throughout the cycle. Reducing ε increases the maximum lift, and the lift curve at ε = −10° is mostly above that at ε = 0°, indicating that reducing ε is beneficial for lift generation during the entire cycle. To further explore the impact of σ and ε on aerodynamic forces, numerical simulations were conducted for underlying mechanism analysis.

Based on the above measurement results, when only σ is changed, the lift-to-weight ratio of the flapping wing in a single cycle ranges from −51.3% to 102.7%, and when only ε is changed, the lift-to-weight ratio of the flapping wing in a single cycle ranges from 97.4% to 230.2%. Therefore, the asymmetry of stroking and pitching motions has a great influence on the lift-to-weight ratio. The maximum lift-to-weight ratio generated by the flapping-wing mechanism designed in this paper is much greater than 100%, which meets the lift requirements for the flight of the flapping-wing mechanism, and the aerodynamic force covers a large range, leaving enough margin for a stable flight state.

### 3.2. Influence of Pitch Deviation Angle on Aerodynamic Characteristics

In this section, Cases 1–5 as defined in Table 3 are selected for analysis. By varying the pitch deviation angle σ, the effect of different pitch deviation angles on the asymmetric pitch motion of the flapping wing is studied through flow field analysis. The results are used to assess the impact of the pitch deviation angle on the aerodynamic forces of the asymmetric flapping wing.

Aerodynamic characteristics were analyzed for each experimental group over a steady motion cycle. In Figure 15, *T* = 0–0.5 represents the downstroke phase, and *T* = 0.5–1 represents the upstroke phase. During the transition of the flapping wing from the stationary state to the start of the downstroke, sudden changes in the flow field at *T* = 0 cause an aerodynamic force spike. Under conditions with consistent motion laws, the aerodynamic characteristics of flapping wings with different pitch deviation angles are compared. Over an entire motion cycle, during the downstroke phase, the lift coefficient is largest at σ = 0°, while the thrust coefficient is smallest at σ = −10°. The remaining three sets of lift coefficients are all smaller than the lift coefficients produced when σ is 0°, at which σ is negatively beneficial to lift when a is not zero; In the upstroke phase, the lift coefficient produced at σ = −10° is the largest, the lift coefficient produced at σ = 10° is the smallest, and the remaining three sets of lift coefficients, from largest to smallest, are −5°, 5° and 0°, and at this time, σ is a negative value of the positive gain on the lift production. Similarly, the change in the flow field at the moment *T* = 0 leads to an abrupt change in thrust, comparing the thrust characteristics of the flapping wing at different pitch deviation angles with the same law of motion. Over an entire cycle, during the downstroke phase, the thrust coefficient decreases sequentially for σ of 10°, −5°, 5°, 0°, 10°. During the upstroke phase, the thrust coefficients from large to small correspond to σ of 0°, −10°, 5°, −5° and 10°, respectively. Further investigation into the impact of pitch deviation angles on aerodynamic performance is presented in Table 4, showing the average lift coefficients generated during each phase for different pitch deviation angles, and Table 5, which shows the average thrust coefficients for each phase.

For the entire cycle, the average lift coefficient when σ = 0° is 0.1093. Compared to σ = 0°, the lift coefficients at σ = −10°, σ = −5°, σ = 5° and σ = 10° are increased by 457.21%, 44.10%, −191.22%, and −444.56%, respectively. Overall, negative values of σ contribute positively to lift generation, with the highest gain observed at σ = −10°. The average thrust coefficient is largest (3.0707) when σ = 0°. For σ = −10°, σ = −5°, σ = 5°, and σ = 10°, the thrust coefficients decrease by 19.9%, 9.72%, 10.81%, and 11.27%, respectively, compared to σ = 0°. At *T* = 0.1, all five experimental groups generate the maximum lift and minimum thrust, while at *T* = 0.9, the lift difference is largest and the maximum thrust is generated. Therefore, the pressure distribution along the span, from the wing root to the wingtip, at *T* = 0.1 and *T* = 0.9 is analyzed, as shown in Figure 16a,b. When R < 0.5, the pressure difference is small. To analyze the distribution of the two-dimensional vorticity at the section positions of 0.6 R–0.8 R (where R is the wingspan), an increment of 0.1 R is chosen for further analysis.

Due to the pressure difference on the surface of the flapping wing, aerodynamic forces are generated. At *T* = 0.1, which corresponds to the downstroke phase, the pressure difference direction is nearly vertical, which is favorable for lift generation. When σ = 0°, the areas of high and low pressure regions along the wingspan are larger than those in the other experimental groups, and the pressure field on the upper and lower surfaces of the wing remains relatively concentrated. Therefore, the maximum pressure difference is generated, resulting in the largest lift, as shown in Figure 15a. At *T* = 0.9, which is also in the upstroke phase, the pressure difference direction is almost horizontal, which is favorable for thrust generation. Similarly, when σ = 0°, the maximum thrust is generated, corresponding to Figure 15b. In summary, when σ = −10°, the average lift coefficient is the largest and the average thrust coefficient is the smallest, resulting in the maximum lift and minimum thrust.

### 3.3. Influence of Stroke Midpoint Angle on Aerodynamic Characteristics

This section analyzes five cases (Cases 6–10), with parameters set in Adams and XFlow. Flow field analysis is conducted for the flapping wing at different stroke midpoint angles. The results are used to assess the impact of the stroke midpoint angle on the aerodynamic forces of the flapping wing.

When the dragonfly is in motion, under the condition that the stroke amplitude and pitch angle amplitude remain constant, by changing the position of the stroke midpoint angle, the position of the flapping wing relative to the body can be adjusted significantly. This results in different variations in the stroke angle and pitch angle, thus yielding distinct aerodynamic characteristics, as shown in Figure 17. Comparing the lift characteristics of the flapping wing at different stroke midpoint angles reveals the following: during the whole cycle, in the downstroke phase, the lift coefficient increases as the stroke midpoint angle ε decreases; in the downstroke phase, the lift coefficient increases as the stroke midpoint angle decreases. Thus, decreasing ε results in a positive contribution to lift generation. Comparing the thrust characteristics of the flapping wing at different ε reveals the following: during the whole cycle, in the downstroke phase, the thrust coefficients from large to small correspond to ε of 0°, 10°, 5°, −5°, and −10°, which are not much different numerically; in the upstroke phase, the thrust coefficients keep increasing as ε decreases. To further investigate the influence of different ε on aerodynamic forces, Table 6 presents the average lift coefficients at each phase for different ε. To further explore the impact of different ε on thrust, Table 7 shows the average thrust coefficients at each phase for different ε, and Table 7 provides the average thrust coefficients for each phase.

For the entire cycle, the average lift coefficient at ε = 10° is 0.6762. Compared to ε = 10°, the lift coefficients at ε = 5°, 0°, −5°, −10° are increased by 65.56%, 88.06%, 140.42%, and 178.81%, respectively. Overall, decreasing ε leads to a positive contribution to lift generation, with the highest lift gain observed at ε = −10°. The average thrust coefficient at ε = 10° is the largest (2.4834). As ε is −10°, −5°, 5° and 10° in sequence., the thrust coefficient decreases by 9.73%, 18.70%, 28.41%, and 38.58%, compared to that when ε = 0°. Figure 18a shows the comparison of three-dimensional vorticity during the downstroke phase (*T* = 0–0.5) at different ε values. During the downstroke, the main source of lift is the LEV structure attached to the wing’s leading edge. The presence of the LEV creates a low-pressure region on the upper surface of the wing, and this region evolves as the LEV structure develops. For each case, the LEV gradually forms during the downstroke, moving toward the trailing edge and mixing with the TEV. At the end of the downstroke, the LEV detaches from the surface of the wing, which corresponds to the lift variation: the lift gradually increases to a maximum value and then decreases to nearly zero by the end of the downstroke phase. With different values for ε, the relative position and velocity of the flapping wing with respect to the body differ significantly at the same point in the flapping cycle. The pressure difference generated by the formation and development of the LEV affects the aerodynamic forces on the flapping wing, which can be decomposed into lift and thrust. By combining the vorticity intensity shown in the figure and the relative position of the LEV with the flapping wing, the distribution of the lift coefficient curve can be determined. Figure 18b shows the comparison of three-dimensional vorticity during the upstroke phase (*T* = 0.5–1) at different ε values. Similarly, the change in thrust coefficient in Figure 18 corresponds to the distribution of vorticity, and at *T* = 1, the LEV detaches from the surface of the wing, marking the end of the upstroke phase. In conclusion, when ε decreases, the average lift coefficient increases, while the average thrust coefficient decreases.

### 3.4. Influence of Pitch Angle Amplitude on Aerodynamic Characteristics

The asymmetric variation pattern of the pitch angle during dragonfly wing flapping is as follows: when the wing reaches the mid-downstroke phase, the wing is approximately parallel to the horizontal plane and moves downward. This allows the vertical component of the air resistance to be converted into the wing’s lift. When the wing reaches the mid-upstroke phase, the wing is nearly parallel to the flapping plane, effectively reducing the aerodynamic resistance generated during the upstroke and minimizing the damage to lift caused by the vertical component of aerodynamic resistance. Early or delayed pitch results in a smaller vertical force compared to symmetric pitch, and the duration of the pitch has a minimal effect on the aerodynamic forces. Therefore, the magnitude of the pitch angle is a crucial parameter in controlling the aerodynamic forces generated by the dragonfly’s flapping wings. To explore the aerodynamic force variations under asymmetric motion and different pitch angles, Cases 11–13 were set up to investigate the effects of varying pitch angles on the flapping wing’s aerodynamic characteristics.

The aerodynamic characteristics curves under different $\varphi^{\prime }$ values are shown in Figure 19. When only the pitch angle amplitude is changed, the average speed during the flapping motion is positively correlated with the pitch angle amplitude. Throughout the entire cycle, as the pitch angle amplitude increases, the peak values of the lift coefficient and thrust coefficient during both the downstroke and upstroke phases continuously increase. This phenomenon occurs because the flapping speed gradually increases, which, in turn, leads to an increase in the pressure difference between the upper and lower surfaces as the pitch angle amplitude increases. To analyze the speed field at the downstroke phase, *T* = 0.1 is selected for the analysis. Figure 20 shows the velocity distribution contour at different $\varphi^{\prime }$ spanwise positions (0.6 R–0.8 R). It can be seen that as $\varphi^{\prime }$ increases, the high-speed regions on the upper and lower surfaces of the flapping wing expand, causing the peak value of the lift curve to increase. Similarly, during the upstroke phase, due to the increased speed field, the peak value of the thrust coefficient also increases as $\varphi^{\prime }$ increases.

To further investigate the impact of different pitch angle amplitudes on aerodynamic forces, Table 8 shows the average lift coefficients at each phase for different pitch angle amplitudes, and Table 9 presents the average thrust coefficients at each phase for different pitch angle amplitudes.

For the entire cycle, the average lift coefficient at $\varphi^{\prime }$ = 45° is 0.6084. The lift coefficients at $\varphi^{\prime }$ = 55° and $\varphi^{\prime }$ = 65° increase by 0.707% and 43.37%, respectively, compared to $\varphi^{\prime }$ = 45°. Overall, increasing $\varphi^{\prime }$ leads to a positive contribution to lift generation. During the downstroke phase, the lift coefficient decreases slightly as $\varphi^{\prime }$ increases, but the numerical change is small. In the upstroke phase, the lift coefficient increases with $\varphi^{\prime }$, indicating that the main benefit of increasing $\varphi^{\prime }$ for lift generation is the reduction in aerodynamic resistance during the upstroke. The average thrust coefficient at $\varphi^{\prime }$ = 45° is 2.4555. At $\varphi^{\prime }$ = 55° and $\varphi^{\prime }$ = 65°, the thrust coefficients increase by 23.09% and 35.06%, respectively, compared to $\varphi^{\prime }$ = 45°. Therefore, increasing $\varphi^{\prime }$ results in a positive contribution to thrust generation, and the increase principle is the same as in the downstroke phase. Thus, both during the upstroke and downstroke phases, the lift coefficient increases with $\varphi^{\prime }$. In summary, adjusting $\varphi^{\prime }$ leads to positive gains in both lift and thrust, resulting in a larger aerodynamic force coverage generated by the dragonfly’s motion.

## 4. Conclusions

In the present study, a single-motor-driven, two-degrees-of-freedom flapping-wing mechanism was independently designed. The flapping motion pattern of the dragonfly was analyzed, and a “stroke-pitch” coupled motion model for the dragonfly wings was established. The designed mechanism is characterized by its small size, light weight, and rich motion patterns, effectively simulating most of the dragonfly’s motion patterns during flight.

To address the asymmetric motion characteristics of dragonfly flapping, the concepts of pitch deviation angle σ and stroke midpoint angle ε were introduced. Based on the designed flapping mechanism, different variable combinations can be achieved by adjusting the mounting points. To study the effect of a single variable, the flapping wing was treated as a rigid structure. Through kinematic and fluid simulations, the following conclusions were drawn:

The variation in σ induces an asymmetric change in the flapping motion. Under the same motion pattern, when σ is reduced by 10°, the average lift coefficient increases by 457.21%, while the average thrust coefficient decreases by 19.9%. When σ is increased by 10°, the average lift coefficient decreases by 444.56%, and the average thrust coefficient decreases by 19.9%. This phenomenon is attributed to the fact that changing σ alters the distribution and integrity of the pressure field at a given moment. Therefore, changing σ within a certain range can significantly affect the aerodynamic forces.

The variation in ε causes asymmetric changes in the flapping motion. With other parameters constant, reducing ε increases the average lift coefficient and decreases the average thrust coefficient, which improves the dragonfly’s lift characteristics. Increasing ε increases the average thrust coefficient and decreases the average lift coefficient, which improves the dragonfly’s thrust characteristics. The aerodynamic adjustment mechanism is as follows: the change in ε alters the relative position of the wing to the body, which modifies the angle of attack of the wing and thus affects the distribution and evolution of LEV in the dragonfly’s body space, ultimately influencing the aerodynamic forces. Based on the previous analysis, the lift of the flapping wing mainly relies on the low-pressure area generated by the LEV. The low-pressure area of TEV is distributed far from the surface of the flapping wing. Therefore, the contribution of the TEV to the aerodynamic force is relatively small.

The variation in $\varphi^{\prime }$ under asymmetric motion provides a better understanding of the flapping mechanism. Increasing $\varphi^{\prime }$ changes the flow field speed, which increases the peak values of both lift and thrust, resulting in a larger aerodynamic force coverage range.

Through experimental assembly and force measurement platform setup for the designed flapping-wing mechanism, the experimental results showed good agreement with the simulation results, validating the scientific basis of the design. This provides theoretical support for the future development of flapping-wing mechanisms and biomimetic flapping-wing aircraft. This article carries out a series of studies on the rigidity of wings and draws some conclusions. However, real wings are flexible, and this article ignored the flexibility of wings when studying the motion of flapping wings. Peng et al. carried out a study on the effect of wing flexibility on aerodynamic forces through numerical simulation, and the results of the study showed that the presence of flexible deformation would have an effect on aerodynamic performance, but sometimes for good and sometimes for bad, and there is no absolute meaning of favorable or unfavorable [21]. In this study, the effect of flexible wing deformation on aerodynamic force was negligible compared with the effect of wing motion law on aerodynamic force; therefore, this paper adopts the rigid wing to carry out numerical simulation, and subsequent research will consider the flexible deformation of the wing.

Real dragonflies have four wings. There is a wingtip interference effect for the different motion patterns of the front and back wings, and a previous study by Zheng et al. found that real dragonflies can utilize wing interaction for aerodynamic force regulation, and that aerodynamic force generation can be improved when the front and back wings are synchronized to downstroke [28]. The aerodynamic implications of the wing interaction coupled with asymmetricity of wing motion will be an interesting topic for future studies using mechanisms for multiple wings.

## Figures and Tables

**Figure 1 biomimetics-10-00309-f001:**
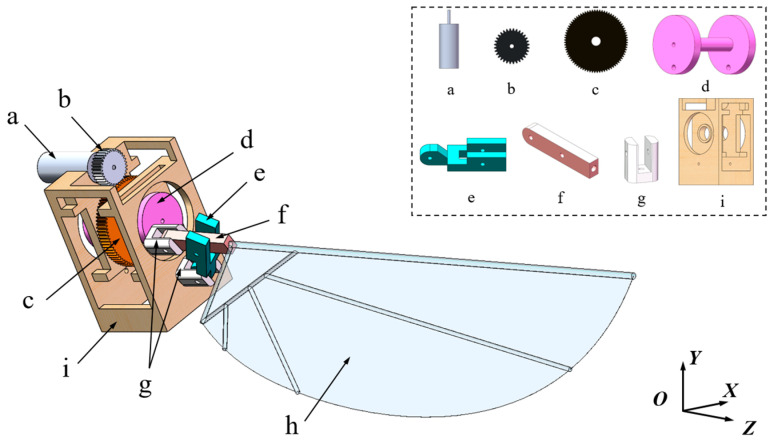
Exploded view of the flapping-wing mechanism. a—Motor; b—Driving gear; c—Driven gear; d—Transmission shaft; e—Linkage 1; f—Linkage 2; g—Rotational axis; h—Flapping wing; i—Frame.

**Figure 2 biomimetics-10-00309-f002:**
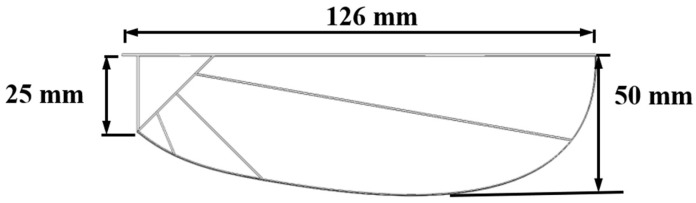
Structure of the flapping-wing model.

**Figure 3 biomimetics-10-00309-f003:**
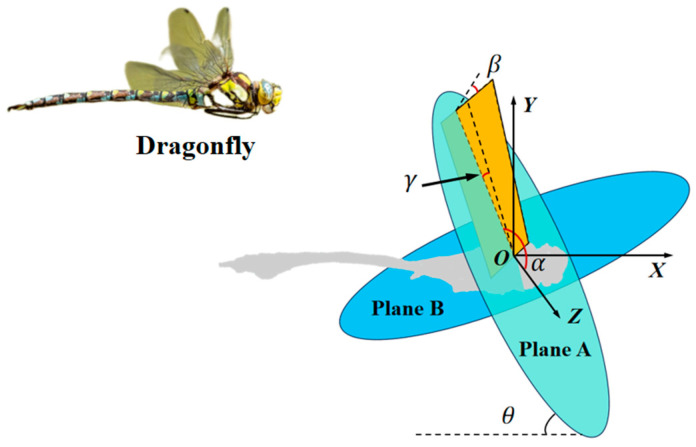
Definition of dragonfly wing model parameters.

**Figure 4 biomimetics-10-00309-f004:**
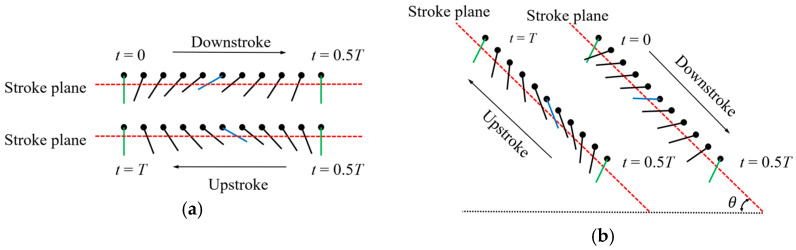
Flapping motion cross-section: (**a**) symmetric rotation; (**b**) asymmetric rotation. The green lines represent the starting and ending positions, and the black lines represent the positions during the movement process.

**Figure 5 biomimetics-10-00309-f005:**
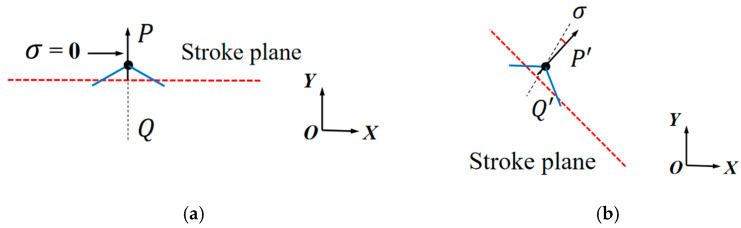
Deviation angle of rotation: (**a**) symmetric rotation; (**b**) asymmetric rotation. The blue line represents the starting and ending positions, and the black line represents the angle bisector.

**Figure 6 biomimetics-10-00309-f006:**
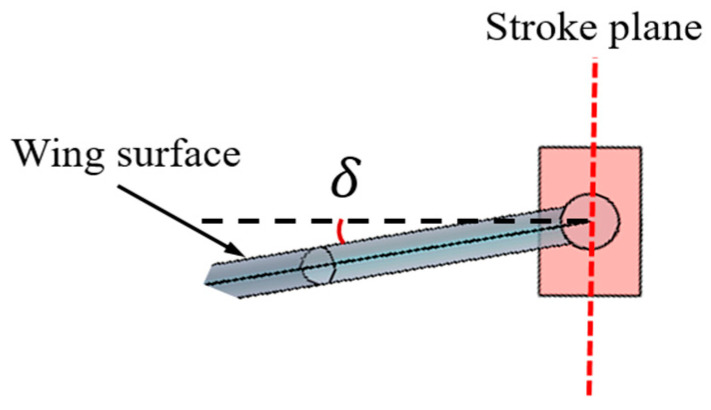
Determination of the pitch deviation angle.

**Figure 7 biomimetics-10-00309-f007:**
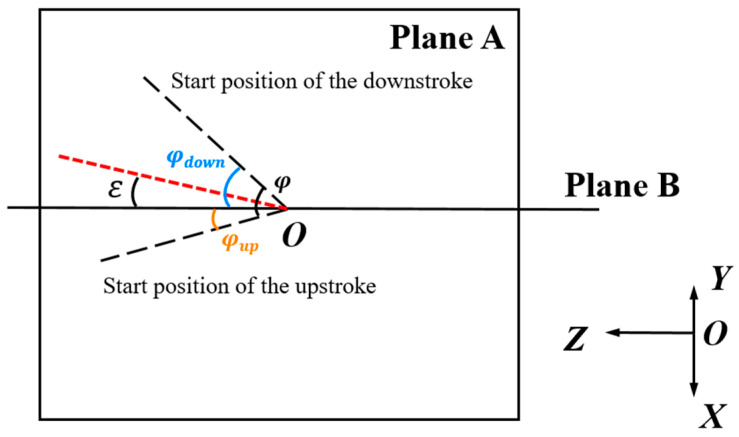
Stroke midpoint angle diagram.

**Figure 8 biomimetics-10-00309-f008:**
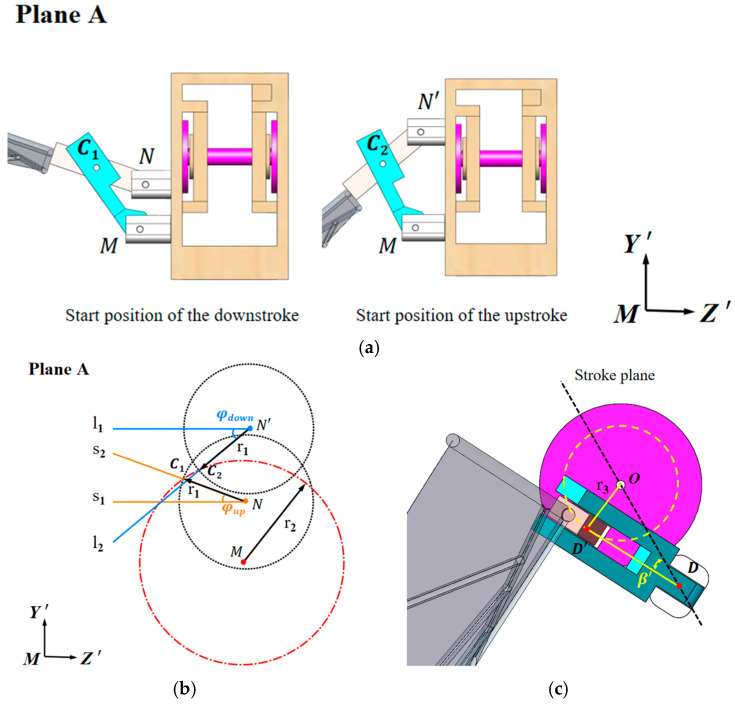
Determination of the stroke midpoint angle and pitch angle extremes: (**a**) Plane A perspective with coordinate definitions for installation points, selecting the extreme positions of the downstroke and upstroke starting points; (**b**) determination of parameters for the two extreme positions and geometric principles; (**c**) determination of pitch angle extremes.

**Figure 9 biomimetics-10-00309-f009:**
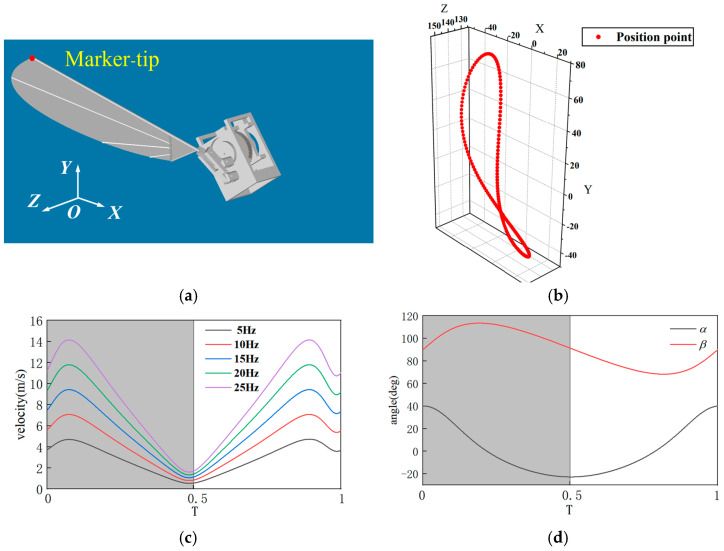
Kinematic analysis of the flapping wingtip: (**a**) position of the wingtip marker point in Adams; (**b**) wingtip motion trajectory in three-dimensional space, approximately “8”-shaped; (**c**) velocity variation curves of the wingtip at different flapping frequencies; (**d**) variation curves of the stroke angle and pitch angle.

**Figure 10 biomimetics-10-00309-f010:**
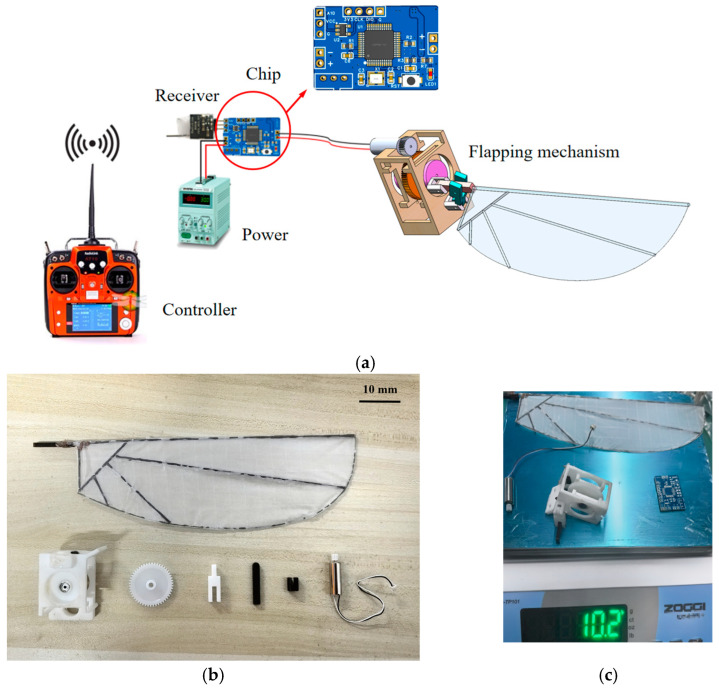
Control principle and physical display: (**a**) motor control principle: self-developed control chip adapted to the current and power requirements of the receiver and motor; (**b**) dimensional display of key components: all parts, except the flapping wing, have a maximum size of 30 mm; (**c**) total weight measurement: the total weight, including the control chip, is 10.2 g.

**Figure 11 biomimetics-10-00309-f011:**
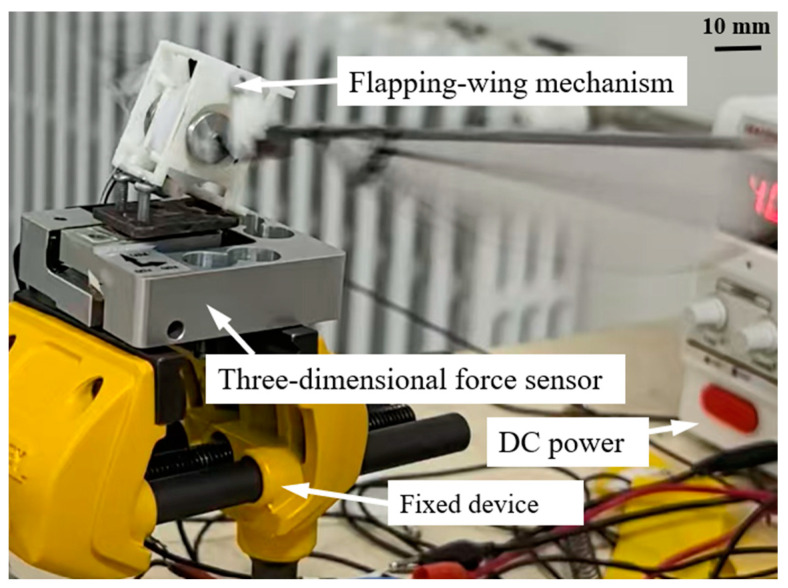
Force measuring platform.

**Figure 12 biomimetics-10-00309-f012:**
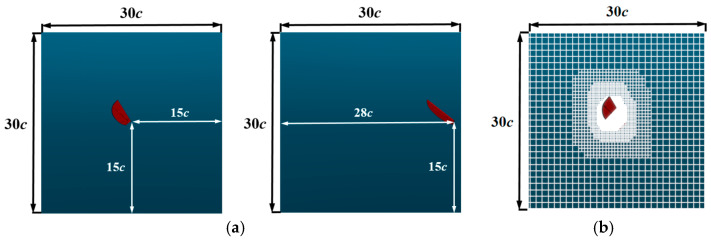
Computational domain initialization setup: (**a**) computational domain range and the relative position of the wing root within the computational domain; (**b**) fluid domain mesh, verifying mesh independence.

**Figure 13 biomimetics-10-00309-f013:**
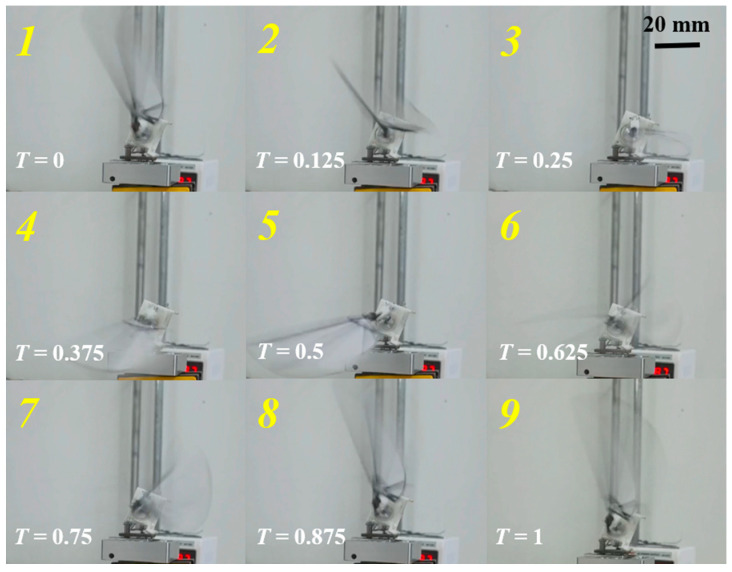
Flapping-wing motion driven by the proposed mechanism within a flapping cycle.

**Figure 14 biomimetics-10-00309-f014:**
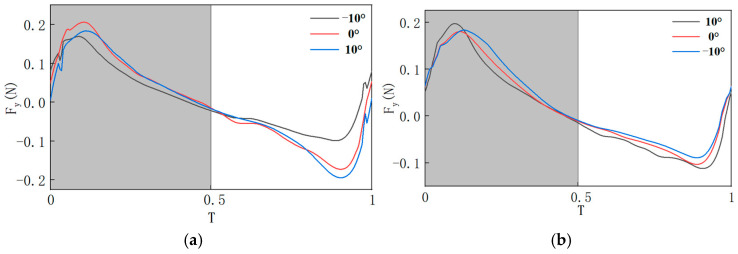
Experimental results of lift under asymmetric motion: (**a**) lift curves at different σ values; (**b**) lift curves at different ε values.

**Figure 15 biomimetics-10-00309-f015:**
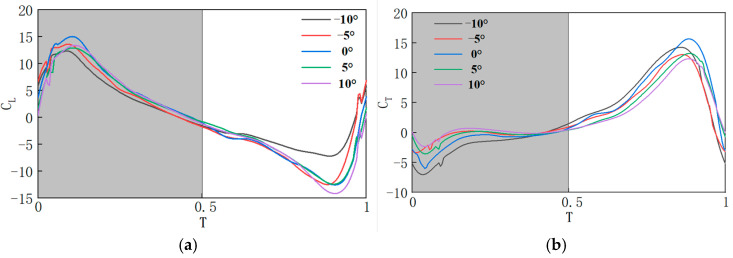
Aerodynamic characteristics curves at different σ values: (**a**) lift coefficient curve; (**b**) thrust coefficient curve.

**Figure 16 biomimetics-10-00309-f016:**
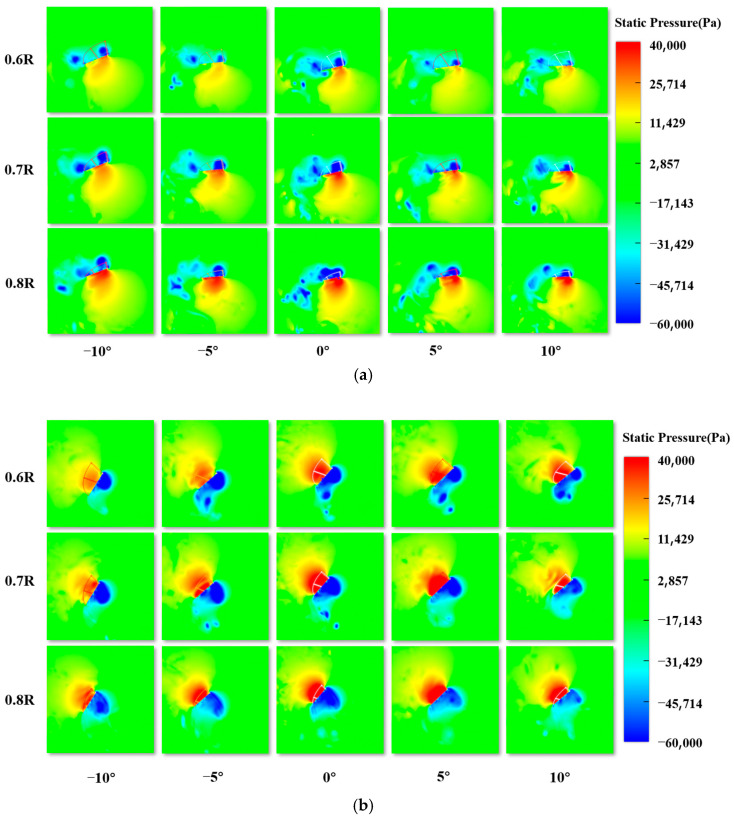
Pressure distribution along the wingspan for different values of σ: (**a**) pressure distribution at *T* = 0.1; (**b**) pressure distribution at *T* = 0.9.

**Figure 17 biomimetics-10-00309-f017:**
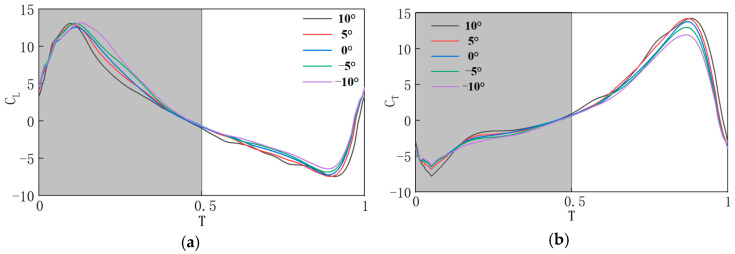
Aerodynamic characteristics curves at different ε values: (**a**) lift coefficient curve; (**b**) thrust coefficient curve.

**Figure 18 biomimetics-10-00309-f018:**
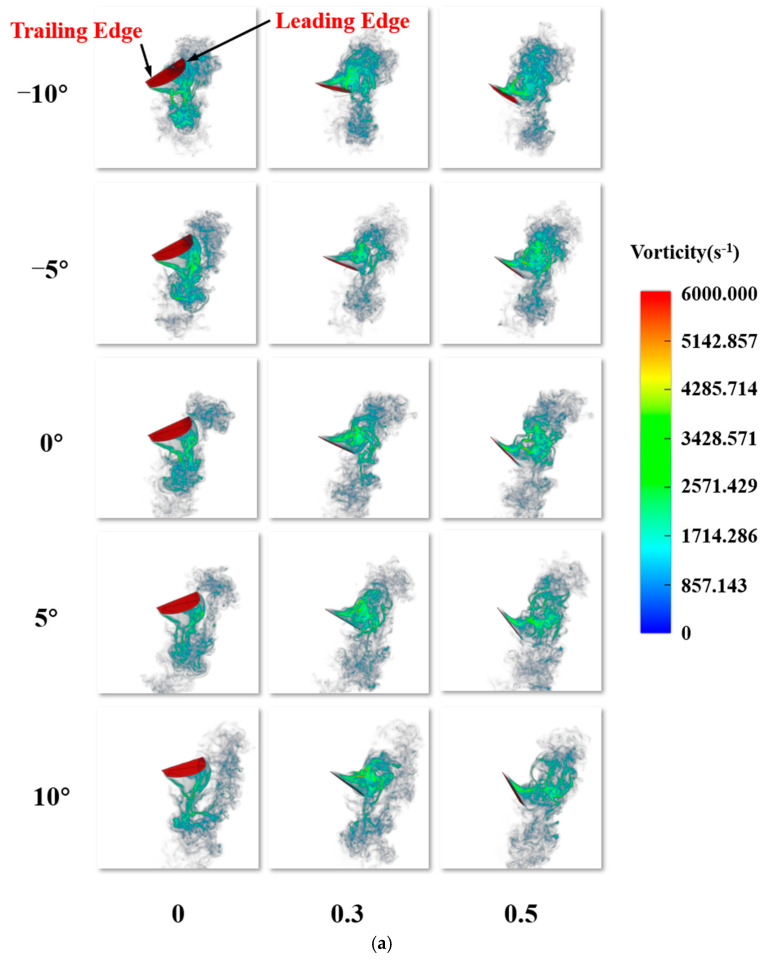
Evolution of three-dimensional vorticity on the flapping wing surface at different ε values: (**a**) wing position and vorticity evolution during the downstroke phase; (**b**) wing position and vorticity evolution during the upstroke phase.

**Figure 19 biomimetics-10-00309-f019:**
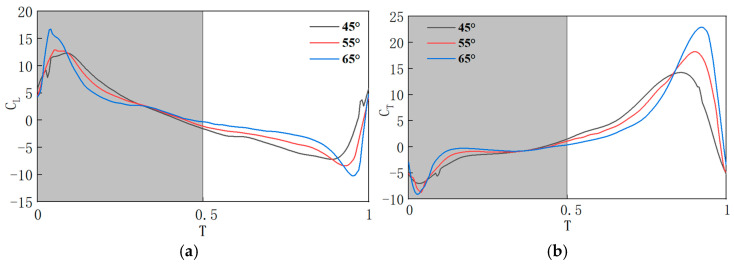
Aerodynamic characteristics curves at different $\varphi^{\prime }$ values: (**a**) lift coefficient curve; (**b**) thrust coefficient curve.

**Figure 20 biomimetics-10-00309-f020:**
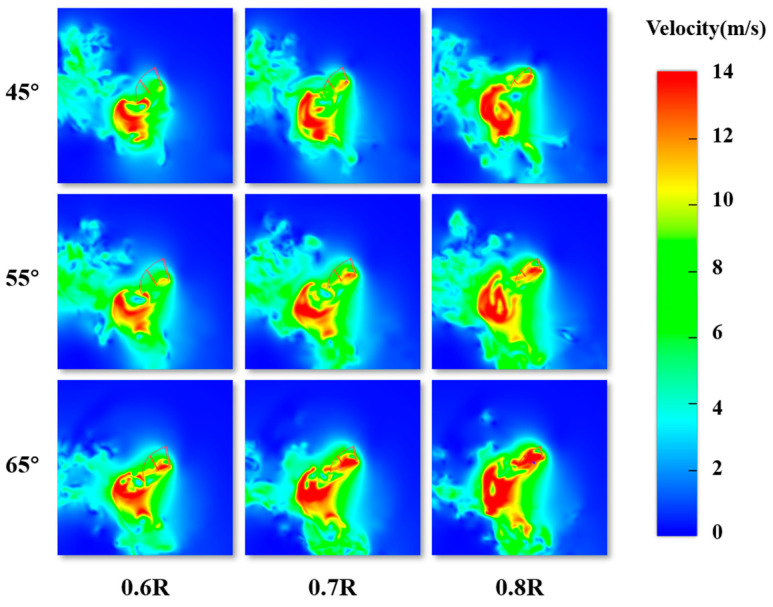
Velocity distribution of the flapping wing along the spreading direction for different $\varphi^{\prime }$ at the moment *T* = 0.1.

**Table 1 biomimetics-10-00309-t001:** Relationship between force and output voltage.

Standard Force (N)	Loaded Output (V)	Unloaded Output (V)	Theoretical Output (V)
0	−0.001	0.000	0.000
2.0	−0.501	−0.500	−0.500
5.0	−1.251	−1.250	−1.250
10.0	−2.502	−2.501	−2.500
15.0	−3.753	−3.752	−3.750
20.0	−5.005	−5.005	−5.000

**Table 2 biomimetics-10-00309-t002:** Experimental group settings.

Group	θ	φ	σ	ε	$\phi^{\prime }$
1	60°	60°	−10°, 0°, 10°	10°	45°
2	60°	60°	0°	−10°, 0°, 10°	45°

**Table 3 biomimetics-10-00309-t003:** Parameter settings for the benchmark cases.

Case	σ	Other Parameters	ε	$\beta_{min}$	$\beta_{max}$	$r_{1}$ (mm)	$r_{2}$ (mm)	$r_{3}$ (mm)
1	10°	θ=60° φ=60° f = 20 Hz	10°	57.5°	102.5°	17.1	11.2	5
2	5°		10°	57.5°	102.5°	17.1	11.2	5
3	0°		10°	57.5°	102.5°	17.1	11.2	5
4	−5°		10°	57.5°	102.5°	17.1	11.2	5
5	−10°		10°	57.5°	102.5°	17.1	11.2	5
6	−10°		10°	57.5°	102.5°	17.1	11.2	5
7	−10°		5°	57.5°	102.5°	15.5	10.3	5
8	−10°		0°	57.5°	102.5°	15.1	10.2	5
9	−10°		−5°	57.5°	102.5°	14.2	9.5	5
10	−10°		−10°	57.5°	102.5°	13.5	9.3	5
11	−10°		10°	57.5°	102.5°	11.2	5	17.1
12	−10°		10°	52.5°	107.5°	10.9	17.1	5.2
13	−10°		10°	47.5°	112.5°	16.7	20.5	7.1

**Table 4 biomimetics-10-00309-t004:** Comparison of average lift coefficients at different pitch deviation angles.

Time	−10°	−5°	0°	5°	10°
*T* = 0~1	0.609	0.1575	0.1093	−0.0997	−0.3766
*T* = 0~0.5	5.158	5.8749	6.6952	5.9875	6.1348
*T* = 0.5~1	−3.961	−5.5596	−6.4881	−6.1941	−6.8962

**Table 5 biomimetics-10-00309-t005:** Comparison of average thrust coefficients at different pitch deviation angles.

Time	−10°	−5°	0°	5°	10°
*T* = 0~1	2.459	2.7721	3.0707	2.7389	2.7247
*T* = 0~0.5	−2.104	−0.4630	−1.3296	−0.6356	−0.1014
*T* = 0.5~1	7.012	5.9866	7.4480	6.0907	5.5288

**Table 6 biomimetics-10-00309-t006:** Comparison of average lift coefficients at different ε.

Time	10°	5°	0°	−5°	−10°
*T* = 0~1	0.6762	1.1195	1.2716	1.6257	1.8853
*T* = 0~0.5	5.6295	6.1334	6.1378	6.6568	6.9328
*T* = 0.5~1	−4.2933	−3.9135	−3.6150	−3.4276	−3.1876

**Table 7 biomimetics-10-00309-t007:** Comparison of average thrust coefficients at different ε.

Time	10°	5°	0°	−5°	−10°
*T* = 0~1	2.4834	2.2417	2.0190	1.7780	1.5253
*T* = 0~0.5	−2.4229	−2.4547	−2.3925	−2.5831	−2.6776
*T* = 0.5~1	7.3746	6.9232	6.4180	6.1277	5.7198

**Table 8 biomimetics-10-00309-t008:** Comparison of average lift coefficients at different pitch angle amplitudes.

Time	45°	55°	65°
*T* = 0~1	0.6084	0.6127	0.8723
*T* = 0~0.5	5.1518	4.995	4.8895
*T* = 0.5~1	−3.9565	−3.8068	−3.1565

**Table 9 biomimetics-10-00309-t009:** Comparison of average thrust coefficients at different pitch angle amplitudes.

Time	45°	55°	65°
*T* = 0~1	2.4555	3.0224	3.3164
*T* = 0.5~1	7.0033	7.8641	8.1699
*T* = 0~0.5	−2.1015	−1.8385	−1.5662

## Data Availability

Available upon request to interested researchers.

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
