# Peer review of "Development of a Dragonfly-Inspired High Aerodynamic Force Flapping-Wing Mechanism Using Asymmetric Wing Flapping Motion"

_biomimetics, 2025, doi:10.3390/biomimetics10050309_

Round 1
Reviewer 1 Report
Comments and Suggestions for Authors
Review report for the manuscript biomimetics-3611903
The flight mechanism designed in this study is inspired by the dragonfly and aims to address asymmetric motion caused by pitch deviation angle and stroke midpoint angle. The experimental results of the mechanism are studied through both kinematic and fluid dynamic simulations. Additionally, this study investigates the variations in aerodynamic forces under asymmetric motion for varied pitch angles. By integrating a bio-inspired mechanism with simulation-based verification, the research results demonstrate innovative concepts and offer valuable insights for the future design of dragonfly-mimicking systems. It is recommended that the paper be accepted for publication after the following comments are properly responded.
- Line 41: “In addition to stroking and pitching, dragonflies also perform other motions. Similarly, other insects may also incorporate additional wing motions during flapping.” Please elaborate the statement more precisely.
- Lines 136-138 and Figure 2: The shape of the wings is crucial, and the materials used to construct the wings will also significantly affect the generation of aerodynamic forces. Please provide sufficient information. The dimensions should also be clearly specified. Were the wing veins designed based on the real wings?
- Lines 167-169: The deviation angle of rotation and pitch deviation angle use the same symbol.
- Line 251: The durations of the upstroke and the downstroke are equal. Is that the normal situation of the dragonfly?
- Lines 258-267: Why does the figure-8 wingbeat pattern exhibit good biomimetic performance? It could be a special pattern in the real flight of dragonfly. Please clarify the statement.
- In addition to verifying the Reynolds number, the Strouhal number should also be calculated as a basis for validation.
- Line 273: Figures 9 (c) and 9(d) are the same. Figure 9(d) does not show the variation curves of the stroke angle and pitch angle.
- Line 324: What is 1/264=10-20%?
- Lines 328-331: Is there any experimental result or relevant literature supporting that this level of error meets the experimental requirements? More information needs to be provided and addressed to enhance the credibility of the results.
- Lines 348-352: What is the rationale for selecting a 10° difference for the pitch deviation angle (σ) and stroke midpoint angle (ε) as the experimental parameters? How might larger or smaller deviations affect the aerodynamic performance? The alignment of the minus sign in Table 2 is incorrect.
- For Figure 13, it would be helpful to include normalized time and the trajectory in each figure for better clarity. Also, if possible, including a video of the flapping mechanism in the appendix would enhance the presentation.
- Figures 16 and 18 are unclear due to low resoluti To improve the quality, please use vector graphics if possible and ensure any raster images should be at least 300 ppi.
This paper also contains some writing errors, as shown below:
- Please check the in-text citation formatting after reference [22], as the square brackets seem to be missing.
- The unit of “Pa” in lines 382 and 383 is misaligned.
3. Line 419: “Section 3.1-3.4” should be “Sections 3.2–3.4”?
4. Line 468: mentions that the thrust coefficient is the largest; however, based on Figure 15 (b), it appears to be the smallest during the downstroke phase.
5. Line 506: The description at T = 0.9 indicates the downstroke phase. However, based on the motion pattern, shouldn’t this actually correspond to the upstroke phase?

Reviewer 2 Report
Comments and Suggestions for Authors
1. The dragonfly has four wings (two pairs), but this article studied a single wing . Can it reveal the high aerodynamic force flapping wing mechanism of dragonfly?
2. In this study, the flapping wing is treated as a rigid structure; In real world, the wings are flexible. What problems may this cause? Has any researcher studied the aerodynamic characteristics of flexible wings? I Suggest that the author to provide necessary explanations in the introduction section.
3. For Figure 18, it is recommended to provide some locally enlarged details to better observe the flow field characteristics and vortex distributions.
4. In the conclusion, only the conclusion about the leading edge vortex is given, and no conclusion or information about the trailing edge vortex is provided.
Author Response
Dear reviewer,
Thank you very much for the comments regarding to our manuscript (No. biomimetics- 3611903). All the precious comments have been considered carefully and the authors have tried their best to revise the manuscript according to the comments. Detailed responses to each comment are presented as follows:
- The dragonfly has four wings (two pairs), but this article studied a single wing . Can it reveal the high aerodynamic force flapping wing mechanism of dragonfly?
Response:
Thank you for this comment. According to this suggestion, explanations have been added in the introduction part. Please see from Line 685:
Real dragonflies have four wings. There is a wingtip interference effect for the different motion patterns of the front and back wings, and a previous study by Zheng et al. [23] found that real dragonflies can utilize wing interaction for aerodynamic force regulation, and that aerodynamic force generation can be improved when the front and back wings are synchronized to downstroke. The aerodynamic implications of the wing interaction coupled with asymmetricity of wing motion of will be an interesting topic for future studies using mechanisms for multiple wings.
[23] Zheng, M.; Peng, L.; Su, G.; Pan, T.; Li, Q. Experimental Investigation on Aerodynamic Performance of Inclined Hovering with Asymmetric Wing Rotation. Biomimetics 2024, 9, doi:10.3390/biomimetics9040225.
- In this study, the flapping wing is treated as a rigid structure; In real world, the wings are flexible. What problems may this cause? Has any researcher studied the aerodynamic characteristics of flexible wings? I Suggest that the author to provide necessary explanations in the introduction section.
Response:
Thank you for this comment. This description has been supplemented in the article. Please see from Line 673:
This article carries out a series of studies on the rigidity of wings and draws some conclusions. However, as you said, real wings are flexible, and this article ignored the flexibility of wings when studying the motion of flapping wings. Peng [21] carried out a study on the effect of wing flexibility on aerodynamic forces through numerical simulation, and the results of the study showed that the presence of flexible deformation would have an effect on aerodynamic performance, but sometimes for good and sometimes for bad, and there is no absolute meaning of favorable or unfavorable. In this article, the effect of flexible wing deformation on aerodynamic force is negligible compared with the effect of wing motion law on aerodynamic force, therefore, this paper adopts the rigid wing to carry out numerical simulation, and the subsequent research will continue to carry out the research on the flexible deformation of the wing.
[21] Peng, L.; Pan, T.; Zheng, M.; Su, G.; Li, Q. The spatial-temporal effects of wing flexibility on aerodynamic performance of the flapping wing. Physics of Fluids 2023, 35, doi:10.1063/5.0136024.
- For Figure 18, it is recommended to provide some locally enlarged details to better observe the flow field characteristics and vortex distributions.
Response:
Thank you for this comment.
The picture has been modified according to your suggestion, the clarity of the picture has been enhanced to facilitate a better observation of the structure of the vortex,please see Figures 18:
(a) |
|
(b) |
Figure 18. Evolution of three-dimensional vorticity on the flapping wing surface at different values: (a) wing position and vorticity evolution during the downstroke phase; (b) wing position and vorticity evolution during the upstroke phase.
- In the conclusion, only the conclusion about the leading edge vortex is given, and no conclusion or information about the trailing edge vortex is provided.
Response:
Thank you for this comment. According to this suggestion, explanations have been added in the conclusion part. Please see from Line 661:
Based on the previous analysis, the lift of the flapping wing mainly relies on the low-pressure area generated by the LEV. The low-pressure area of the trailing edge TEV is distributed far from the surface of the flapping wing. Therefore, the contribution of the TEV to the aerodynamic force is relatively small.
- The English could be improved to more clearly express the research.
Response:
Thank you for this comment. According to your suggestions, national speakers were invited to revise the language and presentation of the article, and improved language descriptions are highlighted in red font in the article, for example:
Sentence 1:
Before: In nature, insect flapping is a coupling of two movements: stroking and pitching. Researchers have identified the pitch mechanism as an important unsteady mechanism for generating high lift in flapping flight.
After: In nature, the flapping movement of insects can be described by three movements: stroke, pitch and deviate. The stroke and pitch movements are mainly related to the cause of aerodynamic force, while the deviate movement is mainly related to the distribution of aerodynamic torque.
Sentence 2:
Before: The "8"-shaped flapping of dragonfly wings is a sophisticated adaptive strategy formed over billions of years of evolution. Through vortex control, energy recovery and movement flexibility, it perfectly balances lift demand and energy consumption. This mechanism not only reveals the physical nature of biological flight, but also provides important inspiration for human engineering.
After: The "8"-shaped flapping of dragonfly wings is a sophisticated adaptive strategy formed over billions of years of evolution. Du and Sun showed that dragonflies follow an "8" shaped movement pattern. During the downstroke stage of their wings, the movement of the flapping wings is almost parallel to the direction of gravity, which can generate a large lift force to overcome gravity and fly. In the upstroke stage, the movement of the flapping wings is almost parallel to the horizontal plane, which can generate a large thrust force to make them fly forward.
Sentence 3:
Before: Most advanced flapping-wing mechanisms currently utilize DC motors to drive flapping motion in a two-dimensional plane, making it challenging to achieve simultaneous stroke and pitch motion. To address this issue, this study proposes a novel transmission mechanism that uses a single motor to enable the coupling of stroking and pitching motions. To meet the requirements for balance and assembly, the flapping-wing mechanism is designed symmetrically. The right-side transmission mechanism is selected for decomposition, as shown in Figure 1.
After: Most advanced flapping-wing mechanisms currently utilize DC motors to drive flapping motion, making it challenging to achieve control on stroke and pitch motion simultaneously. To address this issue, this study proposes a novel transmission mechanism that uses a single motor to enable the coupling of stroking and pitching motions. The mechanism for a right wing is selected for decomposition, as shown in Figure 1.

Reviewer 3 Report
Comments and Suggestions for Authors
The article is devoted to the study of the features of the use of the wing properties in the flight of the dragonfly, as well as the asymmetrical movement of these wings. The article is mainly aimed at borrowing these features in order to use the extraordinary flying skills of the dragonfly in the creation of unmanned aerial vehicles that implement the same principle of flight. The article explores and develops the principle of asymmetrical movement of the flapping wing. At a good engineering level, the development of a flapping wing model was carried out, which was then studied in practice. The description of this design is quite clear, the experiments are also described quite clearly, the results of the study are presented in a visual and understandable form. The lifting force created by the wing is also studied.
It would certainly be interesting for the reader if two wings were made in the developed model, and in such a model the action of these wings would be studied when analyzing the flight of a robot inspired by the flight of a dragonfly. I would also like to remind the authors of the article that the dragonfly has not two wings, but four, and, apparently, the virtuoso flight qualities of the dragonfly are achieved by using the coordinated actions of all four wings together. The use of one wing is only a small step in the direction of research, which cannot and is not capable of realizing even in their entirety the principles and advantages of the dragonfly's flight. One wing of a dragonfly can as little demonstrate these capabilities as one model of a leg cannot demonstrate the running of a horse, and one wheel cannot ensure the movement of a modern car. At the very least, it would be useful if in the conclusion to the article the authors shared plans for a more detailed study taking into account the movement of all four wings. It is obvious that one pair of dragonfly wings acts completely differently from the other pair, they have slightly different purposes, and all four wings, that is, both pairs of wings, act completely differently than any one pair of wings could act. It is recommended that the authors add their opinion in the conclusion to the article, since it is unlikely that a dragonfly with four wings can inspire an attempt to realize a flight with flapping wings using only two wings. If it did, it would be a bit strange.
The bibliography lacks uniformity of form. In many references, the authors' surnames are written in capital letters. Apparently, this is a violation of the form, it should be corrected. It is also somewhat unclear why the letter [J] in square brackets is present in the bibliographic references. Perhaps this means that this article was published in a scientific journal. But such notes are not accepted in the bibliography of articles of this publisher. It should be removed.
Author Response
Dear reviewer,
Thank you very much for the comments regarding to our manuscript (No. biomimetics- 3611903). All the precious comments have been considered carefully and the authors have tried their best to revise the manuscript according to the comments. Detailed responses to each comment are presented as follows:
1、I would also like to remind the authors of the article that the dragonfly has not two wings, but four, and, apparently, the virtuoso flight qualities of the dragonfly are achieved by using the coordinated actions of all four wings together. The use of one wing is only a small step in the direction of research, which cannot and is not capable of realizing even in their entirety the principles and advantages of the dragonfly's flight.
Response:
Thank you for this comment. According to this suggestion, explanations have been added in the introduction part. Please see from Line 685:
Regarding the use of a single wing instead of four wings in this paper's research, your examples and explanations are very vivid. The following are our research ideas and work priorities: Real dragonflies have four wings. There is a wingtip interference effect for the different motion patterns of the front and back wings, and a previous study by Zheng [23] found that real dragonflies can utilize the wingtip interference effect for aerodynamic force regulation, and that aerodynamic force generation can be improved when the front and back wings are synchronized to downbeat. However, the focus of this paper is not on the influence of the wingtip interference effect, but on the basic underlying mechanism of aerodynamic force generation in flapping wings, while the movements of the left and right wings can be regarded as symmetric movements. Therefore, a single wing is chosen to be analyzed in this paper.
[23]Zheng, M.; Peng, L.; Su, G.; Pan, T.; Li, Q. Experimental Investigation on Aerodynamic Performance of Inclined Hovering with Asymmetric Wing Rotation. Biomimetics 2024, 9, doi:10.3390/biomimetics9040225.
2、The bibliography lacks uniformity of form. In many references, the authors' surnames are written in capital letters. Apparently, this is a violation of the form, it should be corrected. It is also somewhat unclear why the letter [J] in square brackets is present in the bibliographic references. Perhaps this means that this article was published in a scientific journal. But such notes are not accepted in the bibliography of articles of this publisher. It should be removed.
Response:
Thank you for this comment. The references have been formatted in accordance with the requirements of MDPI, please see from Line 709:
References
- Lei, Y.; Feng, Z.; Ma, C. Aerodynamic Performance of V8 Octorotor MAV with Different Rotor Configurations in Hover. Machines 2023, 11, doi:10.3390/machines11040429.
- Ryu, Y.; Chang, J.W.; Chung, J. Aerodynamic characteristics of flexible wings with leading-edge veins in pitch motions. Aerospace Science and Technology 2019, 86, 558-571, doi:10.1016/j.ast.2019.01.013.
- Dickinson. THE EFFECTS OF WING ROTATION ON UNSTEADY AERODYNAMIC PERFORMANCE AT LOW REYNOLDS NUMBERS. The Journal of experimental biology 1994, 192, 179-206.
- Luo, G.; Du, G.; Sun, M. Effects of Stroke Deviation on Aerodynamic Force Production of a Flapping Wing. Aiaa Journal 2018, 56, 25-35, doi:10.2514/1.J055739.
- Kramer, F.-J.; Boehrnsen, F.; Moser, N.; Schliephake, H. The submental island flap for the treatment of intraoral tumor-related defects: No effect on recurrence rates. Oral Oncology 2015, 51, 668-673, doi:10.1016/j.oraloncology.2015.03.011.
- Dickinson, M.H.; Lehmann, F.O.; Sane, S.P. Wing rotation and the aerodynamic basis of insect flight. Science (New York, N.Y.) 1999, 284, 1954-1960, doi:10.1126/science.284.5422.1954.
- Pornsin-sirirak, T.N.; Tai, Y.C.; Nassef, H.; Ho, C.M. Titanium-alloy MEMS wing technology for a micro aerial vehicle application. Sensors and Actuators a-Physical 2001, 89, 95-103, doi:10.1016/s0924-4247(00)00527-6.
- Phan, H.V.; Park, H.C.; Ieee. Remotely Controlled flight of an Insect-like Tailless Flapping-wing Micro Air Vehicle. In Proceedings of the 12th International Conference on Ubiquitous Robots and Ambient Intelligence (URAI), Goyang, SOUTH KOREA, Oct 28-30, 2015; pp. 315-317.
- Roshanbin, A.; Altartouri, H.; Karasek, M.; Preumont, A. COLIBRI: A hovering flapping twin-wing robot. International Journal of Micro Air Vehicles 2017, 9, 270-282, doi:10.1177/1756829317695563.
- Phillips, N.; Knowles, K.; Bomphrey, R.J. Petiolate wings: effects on the leading-edge vortex in flapping flight. Interface Focus 2017, 7, doi:10.1098/rsfs.2016.0084.
- Phillips, N.; Knowles, K.; Bomphrey, R.J. The effect of aspect ratio on the leading-edge vortex over an insect-like flapping wing. Bioinspiration & Biomimetics 2015, 10, doi:10.1088/1748-3190/10/5/056020.
- Han, J.-S.; Chang, J.W.; Han, J.-H. An aerodynamic model for insect flapping wings in forward flight. Bioinspiration & Biomimetics 2017, 12, doi:10.1088/1748-3190/aa640d.
- Han, J.-S.; Chang, J.W.; Han, J.-H. The advance ratio effect on the lift augmentations of an insect-like flapping wing in forward flight. Journal of Fluid Mechanics 2016, 808, 485-510, doi:10.1017/jfm.2016.629.
- Addo-Akoto, R.; Han, J.-S.; Han, J.-H. Leading-edge curvature effect on aerodynamic performance of flapping wings in hover and forward flight. Bioinspiration & Biomimetics 2024, 19, doi:10.1088/1748-3190/ad5e50.
- Han, J.-S.; Tuan Nguyen, A.; Han, J.-H. Aerodynamic characteristics of flapping wings under steady lateral inflow. Journal of Fluid Mechanics 2019, 870, 735-759, doi:10.1017/jfm.2019.255.
- Han, J.-S.; Breitsamter, C. Aerodynamic investigation on shifted-back vertical stroke plane of flapping wing in forward flight. Bioinspiration & Biomimetics 2021, 16, doi:10.1088/1748-3190/ac305f.
- Usherwood, J.R.; Lehmann, F.-O. Phasing of dragonfly wings can improve aerodynamic efficiency by removing swirl. Journal of the Royal Society Interface 2008, 5, 1303-1307, doi:10.1098/rsif.2008.0124.
- Maybury, W.J.; Lehmann, F.O. The fluid dynamics of flight control by kinematic phase lag variation between two robotic insect wings. Journal of Experimental Biology 2004, 207, 4707-4726, doi:10.1242/jeb.01319.
- Li, Q.; Zheng, M.; Pan, T.; Su, G. Experimental and Numerical Investigation on Dragonfly Wing and Body Motion during Voluntary Take-off. Scientific Reports 2018, 8, doi:10.1038/s41598-018-19237-w.
- Su, G.; Dudley, R.; Pan, T.; Zheng, M.; Peng, L.; Li, Q. Maximum aerodynamic force production by the wandering glider dragonfly (Pantala flavescens, Libellulidae). The Journal of experimental biology 2020, 223, doi:10.1242/jeb.218552.
- Peng, L.; Pan, T.; Zheng, M.; Su, G.; Li, Q. The spatial-temporal effects of wing flexibility on aerodynamic performance of the flapping wing. Physics of Fluids 2023, 35, doi:10.1063/5.0136024.
- Park, H.; Choi, H. Kinematic control of aerodynamic forces on an inclined flapping wing with asymmetric strokes. Bioinspiration & Biomimetics 2012, 7, doi:10.1088/1748-3182/7/1/016008.
- Zheng, M.; Peng, L.; Su, G.; Pan, T.; Li, Q. Experimental Investigation on Aerodynamic Performance of Inclined Hovering with Asymmetric Wing Rotation. Biomimetics 2024, 9, doi:10.3390/biomimetics9040225.
- Stanford, B.K.; Stanford, B.K.; Saellstroem, E.; Ukeiley, L.; Ifju, P.G. Structural dynamics and aerodynamics measurements of biologically inspired flexible flapping wings. Bioinspiration & Biomimetics 2011, 6, doi:10.1088/1748-3182/6/1/016009.
- Willmott, A.P.; Ellington, C.P. The mechanics of flight in the hawkmoth Manduca sexta. I. Kinematics of hovering and forward flight. The Journal of experimental biology 1997, 200, 2705-2722.
- Rajabi, H.; Ghoroubi, N.; Malaki, M.; Darvizeh, A.; Gorb, S.N. Basal Complex and Basal Venation of Odonata Wings: Structural Diversity and Potential Role in the Wing Deformation. Plos One 2016, 11, doi:10.1371/journal.pone.0160610.
- Norberg, U.M.L.; Norberg, R.A. Evolutionary divergence of body size and wing and leg structure in relation to foraging mode in Darwin's Galapagos finches. Biological Journal of the Linnean Society 2023, 140, 240-260, doi:10.1093/biolinnean/blad053.
- Johansson, L.C.; Norberg, U.M.L. Lift-based paddling in diving grebe. Journal of Experimental Biology 2001, 204, 1687-1696.
- Du, G.; Sun, M. Aerodynamic effects of corrugation and deformation in flapping wings of hovering hoverflies. Journal of Theoretical Biology 2012, 300, 19-28, doi:10.1016/j.jtbi.2012.01.010.
- Peng, L.; Zheng, M.; Pan, T.; Su, G.; Li, Q. Tandem-wing interactions on aerodynamic performance inspired by dragonfly hovering. Royal Society Open Science 2021, 8, doi:10.1098/rsos.202275.
- DiLeo, C.; Deng, X. Design of and Experiments on a Dragonfly-Inspired Robot. Advanced Robotics 2009, 23, 1003-1021, doi:10.1163/156855309x443160.
- Wang, H.; Zeng, L.J.; Liu, H.; Yin, C.Y. Measuring wing kinematics, flight trajectory and body attitude during forward flight and turning maneuvers in dragonflies. Journal of Experimental Biology 2003, 206, 745-757, doi:10.1242/jeb.00183.
- Du, G.; Sun, M. Effects of unsteady deformation of flapping wing on its aerodynamic forces. Applied Mathematics and Mechanics-English Edition 2008, 29, 731-743, doi:10.1007/s10483-008-0605-9.
- Wakeling, J.M.; Ellington, C.P. Dragonfly flight. I. Gliding flight and steady-state aerodynamic forces. The Journal of experimental biology 1997, 200, 543-556.
- Cai, Y.; Su, G.; Zhao, J.; Feng, S. The Coupled Wing Morphing of Ornithopters Improves Attitude Control and Agile Flight. Machines 2024, 12, doi:10.3390/machines12070486.
- Delmonte, O.M.; Castagnoli, R.; Calzoni, E.; Notarangelo, L.D. Inborn Errors of Immunity With Immune Dysregulation: From Bench to Bedside. Frontiers in Pediatrics 2019, 7, doi:10.3389/fped.2019.00353.
- Wang, L.; Song, B.; Zhang, M.; Yang, X.; Sun, Z.; Zhang, W. An optimum structural design method for a torque sensor measuring the control moment of the micro flapping-wing robot. Measurement Science and Technology 2024, 35, doi:10.1088/1361-6501/ad7a12.
- Pan, T.; Li, T.; Yan, Z.; Li, Q. Investigation of turbulence-induced disturbances and their evolution to stall onset in a compressor cascade using large eddy simulation. Chinese Journal of Aeronautics 2025, doi:10.1016/j.cja.2025.103491.
- Shao, W.; Zhang, H.; Zeng, P. Design and aerodynamic characteristic analysis of unfolding-bending bionic flapping-wing aircraft. Journal of Mechanical Science and Technology 2022, 36, 2981-2991, doi:10.1007/s12206-022-0530-y.
- Bhardwaj, S.; Dalal, A.; Biswas, G.; Mukherjee, P.P. Analysis of droplet dynamics in a partially obstructed confinement in a three-dimensional channel. Physics of Fluids 2018, 30, doi:10.1063/1.5030738.
- Chavez-Modena, M.; Martinez, J.L.; Cabello, J.A.; Ferrer, E. Simulations of Aerodynamic Separated Flows Using the Lattice Boltzmann Solver XFlow. Energies 2020, 13, doi:10.3390/en13195146.
- Wang, J.; Zhang, C.; Gu, S.; Yang, K.; Li, H.; Lai, Y.; Yurchenko, D. Enhancement of low-speed piezoelectric wind energy harvesting by bluff body shapes: Spindle-like and butterfly-like cross-sections. Aerospace Science and Technology 2020, 103, doi:10.1016/j.ast.2020.105898.
- Shahzad, A.; Tian, F.-B.; Young, J.; Lai, J.C.S. Effects of flexibility on the hovering performance of flapping wings with different shapes and aspect ratios. Journal of Fluids and Structures 2018, 81, 69-96, doi:10.1016/j.jfluidstructs.2018.04.019.
